# The Effects of Heat Stress on the Transcriptome of Human Cancer Cells: A Meta-Analysis

**DOI:** 10.3390/cancers15010113

**Published:** 2022-12-24

**Authors:** Enzo M. Scutigliani, Fernando Lobo-Cerna, Sergio Mingo Barba, Stephan Scheidegger, Przemek M. Krawczyk

**Affiliations:** 1Department of Medical Biology, Amsterdam University Medical Centers, University of Amsterdam, Meibergdreef 9, 1105AZ Amsterdam, The Netherlands; 2Cancer Center Amsterdam, Cancer Biology and Immunology, Treatment and Quality of Life, 1081HV Amsterdam, The Netherlands; 3ZHAW School of Engineering, University of Applied Sciences, CH 8401 Winterthur, Switzerland; 4Chemistry Department, University of Fribourg, 1700 Fribourg, Switzerland; 5Adolphe Merkle Institute, University of Fribourg, Chemin des Verdiers 4, 1700 Fribourg, Switzerland

**Keywords:** hyperthermia, heat shock response, transcriptome, neoplasms, gene expression, meta-analysis

## Abstract

**Simple Summary:**

When exposed to heat, and other forms of stress, mammalian cells activate a rich and diverse network of processes, pathways and genes which, together, protect them from damage. Even though this process, termed the heat stress response, has been studied extensively for many decades, and is particularly relevant in the context of cancer (treatment), its systemic understanding is still beyond our reach. Here, we explored one aspect of the heat stress response in cancer cells—changes in gene expression—by subjecting 18 datasets to extensive meta-analysis. We found a surprisingly high level of inter-study variability, driven at least in part by the different experimental conditions applied in each study, and an apparent absence of a ‘universal’ gene expression signature. Our results suggest that gene expression changes after heat stress may be largely determined by the experimental context and call for a more extensive, controlled study that examines the effects of key experimental parameters.

**Abstract:**

Hyperthermia is clinically applied cancer treatment in conjunction with radio- and/or chemotherapy, in which the tumor volume is exposed to supraphysiological temperatures. Since cells can effectively counteract the effects of hyperthermia by protective measures that are commonly known as the heat stress response, the identification of cellular processes that are essential for surviving hyperthermia could lead to novel treatment strategies that improve its therapeutic effects. Here, we apply a meta-analytic approach to 18 datasets that capture hyperthermia-induced transcriptome alterations in nine different human cancer cell lines. We find, in line with previous reports, that hyperthermia affects multiple processes, including protein folding, cell cycle, mitosis, and cell death, and additionally uncover expression changes of genes involved in KRAS signaling, inflammatory responses, TNF-a signaling and epithelial-to-mesenchymal transition (EMT). Interestingly, however, we also find a considerable inter-study variability, and an apparent absence of a ‘universal’ heat stress response signature, which is likely caused by the differences in experimental conditions. Our results suggest that gene expression alterations after heat stress are driven, to a large extent, by the experimental context, and call for a more extensive, controlled study that examines the effects of key experimental parameters on global gene expression patterns.

## 1. Introduction

Hyperthermia (i.e., the exposure of tumor tissue to elevated temperatures of 39–44 °C) is gaining interest as an adjunct to radio- and chemotherapy in a wide spectrum of cancers, including cervix, colon, skin and bladder [1], with efficacy supported by an increasing body of clinical trials [2,3,4,5,6]. The therapy-promoting effects of hyperthermia are attributed to various mechanisms, both at the macroscopic and cellular level [7,8,9]. For instance, the elevation of tissue perfusion and oxygenation, in both well- and poorly vascularized regions, are thought to be the main drivers of radio- and chemosensitization by hyperthermia [10,11,12,13,14,15,16,17], as they affect some aspects of the tumor microenvironment that are relevant in disease progression and metastasis, such as hypoxia, acidity and nutrient shortage [18,19]. Moreover, the perfusion-driven increase in oxygen maximizes the cytotoxicity of radiotherapy and different types of chemotherapeutics that show a dependence on oxygen for their efficacy [20]. Accumulating preclinical evidence also suggests that hyperthermia has immunostimulatory effects [21], which might be exploited in novel treatment strategies that include, among others, immune checkpoint inhibition [22]. At the cellular level, heat alters membrane characteristics and induces proteotoxic stress, negatively affecting most compartments and multiple pathways [8].

In response to hyperthermia, as well as to many other forms of stress, cells activate a protective mechanism known as the heat stress response. This process was initially characterized as an activation of a subfamily of chaperone proteins, nowadays known as heat shock proteins [23], and many subsequent studies have confirmed and further explored their role in stress tolerance [24,25,26]. In recent decades, however, Omics-based applications, such as microarrays and proteomics, have been applied to systematically uncover heat stress-induced changes in gene expression in different organisms [27,28,29,30], which led to the consensus that heat stress activates functionally different classes of genes that are involved in proteostasis, metabolism, DNA-repair and detoxification [8]. More recent techniques, such as Precision Run-On sequencing (PRO-seq), which identifies the exact positions of transcribing polymerase II complexes across the genome, have provided detailed additional insights into the transcriptional alterations after heat stress [31,32,33,34]. Furthermore, the role of histone modifications [32,35,36] and key transcriptional regulators in the heat stress response, such as HSF1 [30,33,37], have been assessed by chromatin immunoprecipitation sequencing (ChiP-seq) in different organisms.

Among these various aspects of the heat stress response, gene expression is among the best studied, yielding a considerable body of data that has been mostly generated using different transcriptomics techniques. Some early contributions reported on gene expression changes in organisms such as A. fulgidus, E. coli, S. cerevisiae, C. elegans and A. thaliana [38,39,40,41,42], as well as in a human (lymphoma) cancer cell line [43]. Subsequent studies explored the transcriptome alterations in different cancer cell lines [43,44,45,46,47,48], and found multiple effects in processes related to protein folding, cell cycle, mitosis, and cell death (Table 1). Nevertheless, as the different experiments and studies focus on, and highlight, different processes, pathways or genes in relative isolation, systemic interpretation and systematic understanding of the emerging gene regulation landscape remain challenging. Meta-analysis has, historically, been among highly useful tools in such cases and, indeed, it has been applied over a decade ago by Richter and colleagues to review and compare gene regulation in response to heat stress [8]. The authors of this study relied on a few then-available datasets to reveal the diverse and largely non-overlapping patterns of gene expression after heat stress in different organisms.

Here, we apply a comparable meta-analytic approach, focusing on 16 public and two unpublished datasets that quantified hyperthermia-induced transcriptome alterations in various human cancer cell lines. Our results demonstrate a high degree of inter-study variability in the transcriptome landscape, and an apparent absence of a universal heat stress response signature. This is likely caused, at least in part, by the different experimental conditions adopted (including cell line, heating technique, thermal dose, time after heat stress, experimental and data analysis pipeline), but due to the limited number of datasets, it is not feasible to confirm which parameters are major drivers of variability. Our analysis highlights, therefore, that the results of individual gene expression studies should generally be interpreted in the context of their particular experimental setup, and that extrapolation of these results to other conditions should be exercised with caution. It also calls for a more extensive, controlled study that would examine the effects of some key parameters, notably the cell line, the thermal dose, the heating technique, and the time after heat stress, in direct comparison.

## 2. Materials & Methods

### 2.1. Data Collection and Curation

Embase and PubMed-Medline were systematically searched based on the following MeSH terms and keywords: Hyperthermia, Heat Shock Protein, Heat Shock Response, Genomics, Neoplasms, Transcriptome, Whole Exome Sequencing, Gene Expression Profiling, Microarray Analysis, RNA-Seq, Oligonucleotide Array Sequence Analysis, and Protein Array Analysis. Our search led to a collection of 683 references. 505 references were excluded after reviewing the title and abstract. 172 additional manuscripts were excluded by not meeting the following criteria: (i) reference is a research paper, (ii) full manuscript has been published, (iii) treatment protocols are clearly specified, (iv) appropriate controls are present, (v) raw data is available through a database or upon request, (vi) overall quality. Overall quality assessment considered the following criteria: (i) Publication is peer-reviewed, (ii) Statement of conflict of interest is present, (iii) Study reports on cell lines, experimental conditions and criteria for data quality assessment (iv) Comparison between experimental conditions and control is present. Data extraction, from Gene Expression Omnibus (GEO; https://www.ncbi.nlm.nih.gov/geo/ (accessed on 20 March 2022)), and curation was performed in R, official gene symbols were used to identify the genes. When a microarray dataset contained multiple quantifications of gene expression due to the presence of multiple probes in the microarray format, we selected the probe that reported the highest alteration in gene expression. The datasets were subsequently merged with our unpublished datasets.

### 2.2. Acquisition and Processing of Unpublished Gene Expression Data

HeLa and T24 cells were cultured in at 37 °C at 5% CO_2_ in EMEM (Gibco) and DMEM (Gibco, Hong Kong, China), respectively, supplemented with 10% fetal bovine serum (Gibco), 100 U/mL penicillin (Gibco), 100 U/mL streptomycin (Gibco), and 2 mM L-glutamine (Gibco). 24 h before hyperthermia treatment, cells were seeded in 10 cm culture dishes (Greiner). Hyperthermia (i.e., 42 °C for 1 h) was performed by submerging cell cultures in a calibrated water bath. The incubation was extended by 5 min to compensate for the time required to reach the target temperature. Cells were collected by scraping in ice-cold PBS. RNA isolation and library preparation was performed using the RNeasy kit (Qiagen) followed by a KAPA mRNA Hyperprep (Roche). RNA sequencing was performed on a NovaSeq 6000 in a 150 bp paired ended fashion to a depth of 40 M reads. The reads of HeLa cells were aligned using STAR (v2.7.9a), whereupon post-alignment processing was performed using SAMtools (v1.13). Finally, the mapped reads were assigned to genes using Subread (v2.0.1). Quality control was performed using FastQC (v.0.11.9) and MultiQC (v1.11). For reads derived from T24 cells, unique molecular identifiers (UMIs) were demultiplexed with UMItools (v1.1.2) after post-alignment processing. Further processing was identical to that of HeLa cells. Changes in gene expression between the control and hyperthermia-treated cells were obtained using DESeq2 [50].

### 2.3. Meta-Analysis

All data processing and analysis was performed in R (version 4.1.1). Principal component analysis was conducted using the “prcomp()” function of the built-in “stats” package (https://www.rdocumentation.org/packages/stats/versions/3.6.2/topics/prcomp (accessed on 15 September 2022)). Gene set enrichment and overrepresentation analyses were performed by querying molecular signatures from the Molecular Signatures Database [51,52] using the “msigdbr” package (https://CRAN.R-project.org/package=msigdbr (accessed on 15 September 2022)), followed by testing for enrichment by using the “GSEA()” or “enricher()” function of the “clusterProfiler” package at default settings [53]. Set theory was performed using the “VennDiagram” package (https://CRAN.R-project.org/package=VennDiagram (accessed on 15 September 2022)). Hierarchical clustering was performed using the “pheatmap” package (https://CRAN.R-project.org/package=pheatmap (accessed on 15 September 2022)).

### 2.4. Experimental Validation of Gene Expression Changes

HT-29 cells were cultured in DMEM/F-12 (Gibco). HeLa and MCF-7 cells were cultured in MEM (Gibco). The media contained aforementioned supplementation. 24 h before hyperthermia treatment, cells were seeded in flat bottom 6-well plates (Greiner, Hong Kong, China), at a density of 300,000 cells/well. Cells were harvested by scraping in ice-cold PBS, and RNA was subsequently extracted using the PureLink™ RNA Mini Kit (Thermo Fisher, Waltham, MA, USA). cDNA was synthesized from 1000 ng of RNA with the Thermo Verso cDNA Synthesis Kit (Thermo Fisher). Quantitative PCR was performed using a mixture of 10 μL of SYBR Green I (Molecular Probes Inc. Europe. BV), 1 μL specific primers (10 mM), and 100 ng cDNA on a CFX96™ Real-Time PCR Detection System. The specific forward and reverse primers for CRYAB were TCCAGTCCTTTAAACTGAGAGCTA and CATTCCCATCACCCGTGAAGAG, respectively. GAPDH (forward primer: TGCCCAGTTGAACCAGGCG and reverse primer: CGCGGAGGGAGAGAACAGTGA) was used as a control to quantify the fold change in CRYAB expression. All qPCR experiments were performed at least in triplicate.

## 3. Results

### 3.1. Quantification of Gene Expression after Heat Stress in Bladder and Cervix Cancer Cell Lines

To evaluate gene expression changes under clinically relevant hyperthermia conditions, we subjected HeLa (cervical cancer) and T24 (bladder cancer) cell lines to hyperthermia for 1 h at 42 °C using a water bath and harvested them 6 (T24) or 24 (HeLa) hours after the end of the treatment. Cells were then processed for gene expression analysis by total RNA sequencing using standard protocols.

### 3.2. Gene Expression Patterns Are Independent from Key Experimental Parameters

16 transcriptomic datasets originating from six peer-reviewed publications, and two new datasets from our group, were included in the meta-analysis (Figure 1A). Experiments performed to obtain these datasets widely varied in key parameters: heating temperature and duration, the used cell line, the heating technique, and the time between the end of hyperthermia treatment and the transcriptomic analysis (Table 2). Different heating methods can result in considerably divergent biological responses [54,55]. Modulated electro-hyperthermia, for instance, induces more apoptosis than capacitive or conductive hyperthermia under isothermal conditions, due to specific deposition of energy at the cell membrane that may activate death-related signaling pathways [55]. There are no clear relationships, however, between a heating method and gene expression, and to compare the outcomes of the different treatment protocols, we normalized the thermal dose only—by calculating the cumulative equivalent duration of heating at 43 °C, expressed in minutes (CEM43) [56]. Although other approaches, such as deviation from CEM43, and normalization of fractionated heating (e.g., TRISE), have been leveraged to evaluate and estimate the thermal dose and the treatment efficacy in a clinical settings [6,57,58,59], we are currently limited to CEM43 for in vitro studies.

A global visualization showed that the expression of various genes was altered by at least by 1.5-fold (Figure 1B), and that this was accompanied by a shift of median gene expression in some of the datasets (Figure 1C). To assess similarities between the datasets, a principal component analysis was then performed, based on the expression of 9473 genes shared between the datasets. The first three principal components, which represent 54% of the total variance (Figure 1D), revealed clustering of most datasets, with the exception of those originating from Furusawa et al. (2011). Based on this subset of data, the time point after hyperthermia exposure or CEM43 did not appear to play a major role in the clustering outcomes (Figure 1E). Similar findings were obtained by hierarchical clustering of the datasets based on genes that were found to be differentially expressed in at least one dataset (Figure 1F). We therefore conclude that the changes in gene expression patterns are not primarily driven by key experimental parameters (Table 2). For this reason, we performed some of the subsequent analyses with both the full dataset, and a dataset split into two clusters based on the results of the principal component analysis. Cluster 1 contains the studies of Amaya (2014), Andocs (2015), Court (2017), Scutigliani (2022), Tabuchi (2008) and Yunoki (2016), whereas cluster 2 contains the study of Furusawa (2011). Although we are equally interested in all datasets, we will base our conclusions primarily on the results of the datasets from cluster 1, because cluster 2 contains data from a single study.

### 3.3. Common Patterns in Transcriptome Changes at a Pathway Level

To uncover which cellular processes are affected by hyperthermia, we performed both a gene set enrichment analysis (GSEA) and an overrepresentation analysis per dataset for various gene set collections that are available through the molecular signatures database [51,52]. Using GSEA, an enrichment of multiple hallmark gene sets was found across datasets, and various hallmarks were shared to varying extents between the datasets of cluster 1, including the unfolded protein response, MYC targets, E2F targets, mTORC1 signaling and G2-M transition (Figure 2A,B). In contrast, little overlap in enriched hallmarks was found for cluster 2, with the most shared hallmark (i.e., TNF-a signaling) being enriched in only three datasets. 

In contrast to GSEA, the outcome of an overrepresentation analysis depends solely on differentially expressed genes. Given the low amount of differentially expressed genes for some datasets (Table 2, Figure 1B), we also conducted an overrepresentation analysis to evaluate whether the observed significance in hallmark gene set enrichment is actually driven by the differentially expressed genes. The overrepresentation analysis of hallmark gene sets clearly revealed fewer commonly altered pathways between the datasets of cluster 1, and a high degree of variation (Figure 2C–D). Interestingly, gene sets associated with altered KRAS signaling, inflammatory responses, TNF-a signaling and epithelial-to-mesenchymal transition (EMT) appeared as a shared trait in cluster 2, and as a partially shared feature between the two clusters. 

By performing an overrepresentation analysis of Gene Ontology (GO) biological processes, we were able to validate heat shock-driven alterations in protein folding, cell cycle division and mitosis, and cell death, as claimed by the studies that are included in this meta-analysis (Figure 2E–F, Table 1). Again, this overrepresentation analysis yielded highly variable outcomes in cluster 1, and a small degree of overlap between cluster 1 and 2 (Figure 2E). From these analyses, we conclude that the transcriptome alterations between these datasets share some commonalities, but are highly diverse. Importantly, a universal heat stress response signature appears to be absent.

### 3.4. Shared Transcriptome Changes Are Driven by Highly Variable Gene Expression Patterns

To uncover which genes drive the results of the enrichment and overrepresentation analysis, we applied several analytical methods. We firstly assessed the overlap in differentially expressed genes between datasets for each cluster (i.e., a minimum of 1.5-fold change in expression). Interestingly, although 9,474 genes are shared between all datasets, there were no genes changed in all nine studies assigned to cluster 1. Only one differentially expressed gene (i.e., CRYAB) was shared among eight out of nine datasets. Of the 20,592 genes shared between the datasets in cluster 2, 2834 genes were differentially expressed. To evaluate the contribution of these shared differentially expressed genes to the overall result of the overrepresentation analysis, we selected genes that were differentially expressed in at least three datasets in cluster 1, and at least seven in cluster 2. Under these criteria, we found that 72 differentially expressed genes are shared between the clusters (Figure 3B). An overrepresentation analysis, that used this core gene set as input, revealed an enrichment of several hallmark gene sets. Interestingly, some of these hallmarks were also found to be a shared trait between the clusters based on the overrepresentation analysis, such as those related to KRAS signaling, TNF-a signaling and EMT (Figure 3C). Zooming in on the individual genes that drive this enrichment, however, revealed their highly variable expression across datasets (Figure 3D). No GO terms were found to be enriched in the 72 genes shared between the clusters. Driven by the consensus that the expression of heat-responsive genes, especially chaperone proteins, is a hallmark of the heat stress response, and the fact that the overrepresentation analysis showed a tendency towards an enrichment for terms related to cellular responses to heat (Figure 3C), we examined the expression of individual genes that are assigned to this GO term (Figure 3E). Again, a high variability in gene expression could be observed. We thus conclude that common alterations in the gene sets we described in earlier sections are driven by similar genes, but the expression of these genes is, nevertheless, highly variable.

To obtain extra insights into the patterns of transcriptome alterations at gene resolution, we performed a leading edge analysis that uses the overrepresentation analysis as a base. The leading edge constitutes the set of genes that is most responsible for the significant enrichment result [51]. As a gene can be involved in multiple cellular processes, and thus be assigned to various gene sets, exploring the overlap in the leading edges of genes might lead to the identification of genes that are important drivers of the overall outcome [51]. We performed a leading edge analysis for the complete dataset and separately for each cluster, and included hallmark gene sets that were found to be enriched in at least one dataset (Figure 4A). This led to the identification of genes that contributed to the enrichment of multiple hallmarks in cluster 1 and 2 (Figure 4B). By setting a threshold for genes that contributed to more than two gene sets in cluster 1, and more than four gene sets in cluster 2, we found that eight genes overlapped (Figure 4C), and that these genes are involved in most aforementioned shared hallmark gene sets, namely KRAS signaling, EMT, inflammatory responses, and TNF-a signaling (Figure 4D). Thus, although a high degree of variation occurs at the transcriptome level, particular alterations seem to be shared among part of the datasets.

### 3.5. The Small Heat-Shock Protein CRYAB Is Most Commonly Overexpressed after Hyperthermia

Although we found that a high degree of variability in transcriptome alterations exists between the studies, we set out to validate the differential expression of Crystallin Alpha B (CRYAB), a member of the small heat shock protein (HSP20) family, which was found to be most frequently differentially expressed (Figure 5A,B). Three cancer cell lines (i.e., HeLa, MCF-7, HT-29) that were used for the experiments of Amaya (2014), Andocs (2015), and our unpublished data (Scutigliani 2022b), were subjected to hyperthermia, and transcript expression was evaluated at different timepoints using a quantitative PCR (Figure 5C). A robust induction of CRYAB expression was observed four hours after hyperthermia treatment (Figure 5D). These findings resonate with the consensus that CRYAB is heat-inducible [60], and confirm that our meta-analytic approach can provide actionable information on mechanisms underlying the heat stress response. 

## 4. Discussions

Studies that evaluated global gene expression alterations, or specifically genes regulated by certain transcription factors, have shown that cells, in response to hyperthermia, tune a broad range of pathways that involve thousands of genes [8,27,28,30,31,32,33,34,36,37,61]. Since the publication of a meta-analysis by Richter and colleagues [8], which focused on comparing expression alterations across organisms, multiple studies have reported transcriptomics data of heat-stressed cancer cells [43,44,45,47,48,49]. Here, we leveraged this data, as well as our unpublished results, in an attempt to uncover universal patterns in the heat stress response. 

Our systematic literature search yielded datasets with diverse characteristics of the thermal dose, the used cell line, the heating technique, and the timing of the transcriptomic analysis (Table 2). An initial global analysis of the data (Figure 1) revealed a high degree of variability between the transcriptome alterations reported by the studies, with several thousands of genes being altered in a subset of datasets, and almost no alterations in others. While it is known that the status of the transcriptome can change rapidly after heat stress [31,33], there was no clear relation between the expression profiles and any of the key experimental parameters. It should be noted, however, that, due to the limited number of studies, such analysis was considerably underpowered, and a correlation cannot be excluded. Thus, the high degree of variation and the inability to pinpoint the source of this variability limits the strength of the conclusions that can be drawn from the analysis. 

Through pathway analysis, we could confirm the alteration in expression of genes involved in protein folding, cell cycle, mitosis and cell death, in line with the conclusions drawn by the authors of the studies included in the meta-analysis (Figure 2). For instance, Amaya and colleagues [45] pointed out the changes in cell cycle-related genes, in particular those involved in G2/M transition and mitosis. In addition, perturbations of proteostasis by hyperthermia have been observed in microarray data from the studies of Tabuchi and Furusawa [43,48]. With the exception of global cellular stress, we did not observe, however, transcriptome perturbations that were consistent across all datasets. While GSEA initially revealed shared alterations in pathways that are understudied in the context of hyperthermia, such as various aspects of cellular metabolism (e.g., oxidative phosphorylation and glycolysis), as well as mTOR and MYC signaling [62,63,64,65], an overrepresentation analysis showed less commonalities. Interestingly, despite this high variation, we consistently found an alteration of genes related to KRAS signaling, TNF-a signaling and EMT, which connects with recent preclinical and clinical studies that assessed the role of KRAS signaling in cellular responses to hyperthermia [66,67], and *in vitro* data that shows attenuation of TNF-a signaling [68,69]. It is noteworthy, however, that even the expression of genes that are involved in these pathways varied greatly across the datasets (Figure 3). These inconsistencies also have a negative impact on attempts to pinpoint genes that play a key role in these pathway-level effects by leading edge analysis (Figure 4). In short, although the various types of analyses applied here revealed interesting patterns, the patterns are not universally shared by the analyses, studies or datasets. These results call for a more controlled and comprehensive study that could evaluate the effects of various parameters, most notably the thermal dose, the cell line, and the time after heat exposure, through Omics-based approaches or global genome intervention techniques (i.e., CRISPR KO screening). Such study could not only reveal novel, universal mechanisms driving cellular responses to heat stress, but also potential druggable targets to improve the existing, or develop novel therapies based on hyperthermia.

## Figures and Tables

**Figure 1 cancers-15-00113-f001:**
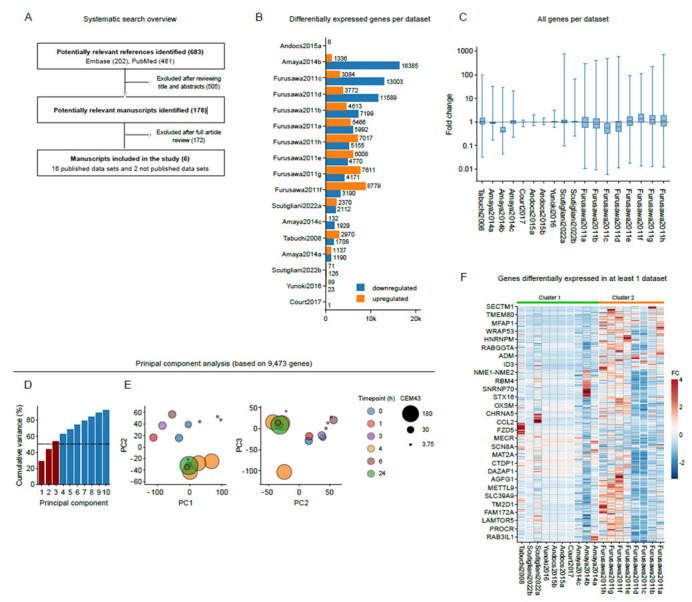
Hierarchical clustering and principal component analysis reveal similarities in gene expression patterns that are independent of experimental characteristics. (**A**) Flow chart displaying the systematic search and selection process for the meta-analysis. (**B**) Number of differentially expressed genes (1.5^−1^ ≥ fold change ≥ 1.5) per study. (**C**) Fold change in gene expression per study. Genes that were shared among all studies were used in the principal component analysis. (**D**) Cumulative histogram of eigenvalues of principal components. The dotted horizontal line marks 50% of the cumulative variance. Principal components visualized in (**E**) are indicated in red. (**E**) Visualization of the principal component analysis. The time point after hyperthermia treatment in hours (h) and the thermal dose, calculated as cumulative equivalent minutes at 43 degrees Celsius (CEM43), are indicated. (**F**) Genes that are differentially expressed in at least one study. Fold change (FC) is centered and scaled per row.

**Figure 2 cancers-15-00113-f002:**
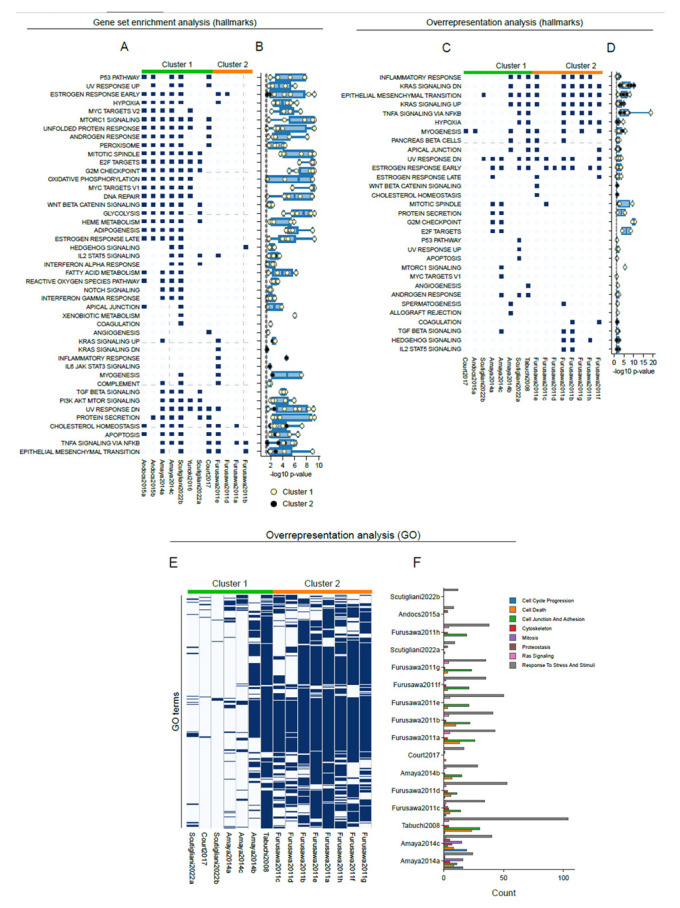
Transcriptome changes in subsets of datasets are highly variable. (**A**) Significantly enriched hallmark gene sets (indicated in blue) per dataset in cluster 1 and 2, as determined by a gene set enrichment analysis. (**B**) Significance of all hallmark gene sets that are displayed in (**A**). Point color indicates the cluster in which a significant enrichment was found. Identical visualizations of an overrepresentation analysis for hallmark gene sets are shown in (**C**) and (**D**). (**E**) Significantly enriched gene ontology (GO) biological process terms, per study, in cluster 1 and 2, as determined by an overrepresentation analysis. (**F**) Hand-curated categorization of GO terms that were found to be enriched per dataset.

**Figure 3 cancers-15-00113-f003:**
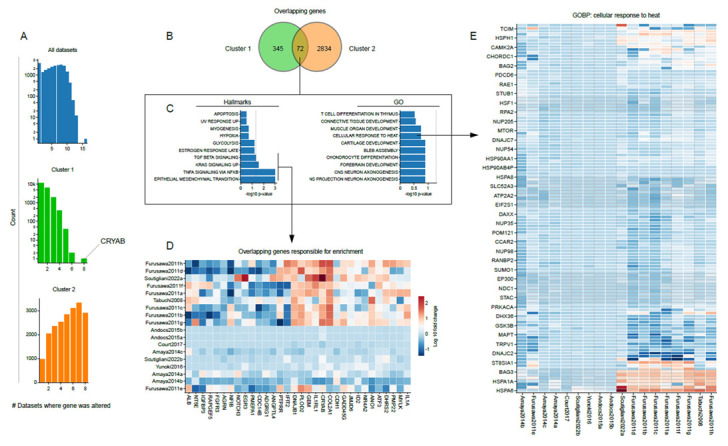
Different genes drive transcriptome changes. (**A**) The frequency at which genes are differentially expressed in the datasets assigned to cluster 1 and 2. (**B**) Venn diagram showing the overlap in genes that were found to be differentially expressed in at least three datasets of cluster 1 and seven datasets of cluster 2. (**C**) The result of an overrepresentation analysis (hallmark and gene ontology (GO) biological processes of gene sets) that used 72 genes shared between the clusters, as shown in (**C**), as input. (**D**) Log fold expression change of the genes that contributed to the enrichment of hallmark gene sets shown in (**C**). (**E**) Log fold expression change of genes that are assigned to the GO term “response to heat stress”. The scaling is identical as in (**D**).

**Figure 4 cancers-15-00113-f004:**
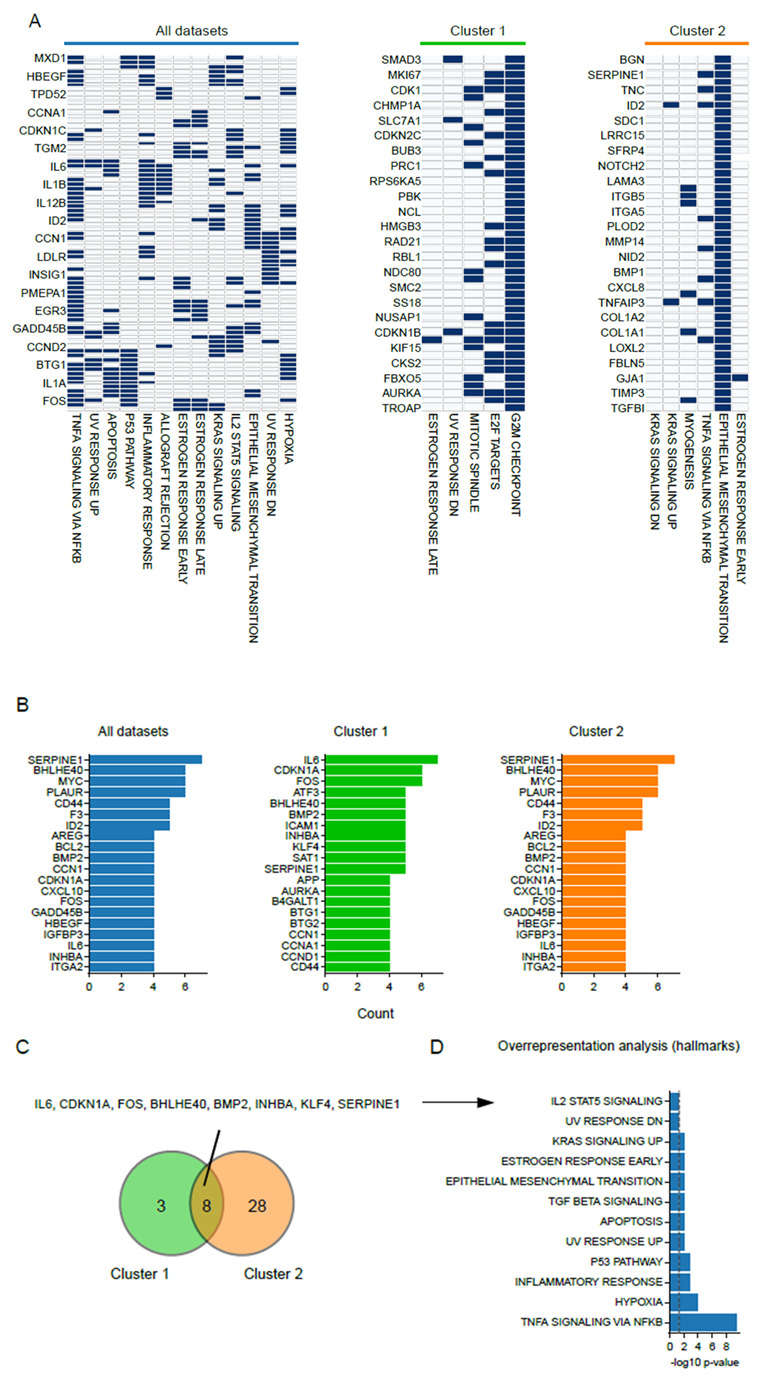
Leading edge analysis reveals commonalities in gene expression profiles. (**A**) Leading edge analysis based on hallmark gene sets that were found to be enriched in the overrepresentation analysis. The analysis was carried out on all datasets, cluster 1, and cluster 2. Genes that appear in a hallmark gene set are marked in blue. (**B**) Top 20 most found genes in the leading edge analysis of all datasets, cluster 1, and cluster 2. (**C**) Overlap in most frequently overexpressed genes that were found at least three times in cluster 1, and 4 times in cluster 2. (**D**) Result of the overrepresentation analysis for hallmark gene sets that used the eight overlapping genes between the clusters, as shown in (C), as input.

**Figure 5 cancers-15-00113-f005:**
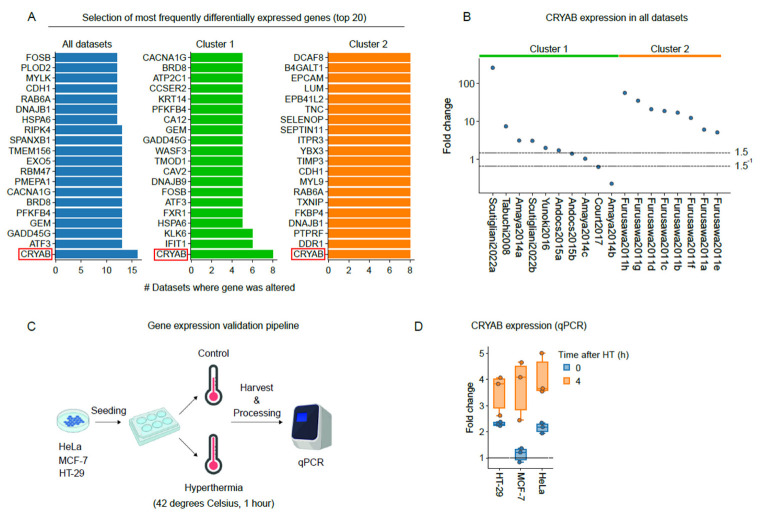
CRYAB expression is enhanced in response to hyperthermia. (**A**) Top 20 most frequently differentially expressed genes in all datasets, and separately in cluster 1 and 2. (**B**) Fold change in CRYAB expression in all datasets. The gray lines depict fold changes that were used as inclusion criteria for differentially expressed in the meta-analysis (i.e., 1.5^−1^ ≥ fold change ≥ 1.5). (**C**) Schematic representation of the pipeline that was used to quantify CRYAB expression using quantitative polymerase chain reaction (qPCR). (**D**) Fold change in CRYAB expression at several time points after hyperthermia (HT) in three cancer cell lines, based on three independent experiments. The horizontal line depicts a fold change of 1.

**Table 1 cancers-15-00113-t001:** Comparison of important pathways and regulated genes between studies.

Study	GEO Accession	Aim	Highlighted Pathways and Functions	Highlighted Genes
Tabuchi et al., 2008 [43]	GSE10043	Examine gene expression patterns in human myelomonocytic lymphoma U937 cells exposed to mild hyperthermia	Up- or downregulated: Cellular function and maintenance Cell cycle Cell death	Unfolded protein response-related genes were upregulated: Hsp40 homologs (DNAJA1, DNAJB1), Hsp70 proteins (HSPA6, HSPA1A, HSPA1B, HSPA1L), HSPB1 (heat shock 27 kDa protein 1), HSPH1 (heat shock 105 kDa/110 kDa protein 1), PPP1R15A (protein phosphatase 1, regulatory (inhibitor) subunit 15 A), and SERPINH1 (serpin peptidase inhibitor, clade H (heat shock protein 47), member 1).
Furusawa et al., 2011 [48]	GSE23405	Understand the molecular mechanisms underlying cellular responses to heat stress at temperatures higher and lower than the inflection point of hyperthermia	Upregulated: Cellular compromise Cellular function and maintenance Cell death Downregulated: Gene expression Cellular growth and proliferation Cellular development	Peak expression of HSPs was observed 3 h after heat stress. The expression level of HSPs such as the Hsp70 (HSPA6, HSPA1A), Hsp40 (DNAJA1, DNAJB1) and Hsp27 (HSPB1) gene subfamilies was gradually elevated at 44 ˚C.
Amaya et al., 2014 [45]	GSE48398	Identify the unique gene networks distinct between normal and cancer cell lines following hyperthermia	Upregulated: Mitosis Cell division Cell cycle	Mitotic regulatory genes were up-regulated: STAG2, NEK2, KPNA4, IPO5, TNPO1, CCNB1, CDK1, CDK6, NCAPG, NCAPG2, TOP2A, NUF2, CENPE, CENPF, ZWILCH, PDS5A, WEE1, KIF11, CHUK, and PPP1CB.
Court et al., 2017 [44]	GSE92990	Investigate gene expression profiles after magnetic fluid hyperthermia in ovarian cancer cell lines to elucidate cellular response and select molecular targets to enhance its effect *in vitro* and *in vivo*	Upregulated: Response to unfolded proteins Response to protein stimulus Protein folding	Top genes related to the aforementioned functions affected by magnetic fluid hyperthermia were HSPs, Hsp70 (HSPA6/HSPA7, HSPA1A, HSPA1B, HSPA1L, HSPA4L), Hsp60 (LOC643300), Hsp40 (DNBAJ family), Hsp20 (CRYAB) and Hsp27 (SERPINH1), and BAG3 (modulator of Hsp70).
Andocs et al. 2015 [49]	GSE58750	Identify the gene expression alterations induced by heat treatment in human tumor HT29 colorectal cancer xenograft mouse model	Upregulated: Heat shock proteins	Members of the heat shock protein 70 family including HSPA1A, HSPA1B, HSPA4, HSPA6, and HSPA8, and their co-chaperones Hsp40 (DNAJB1 and DNAJB4) and Bag3 became upregulated. Hsp90 alpha (HSP90AA1) and Hsp60 (HSPD1) gene transcripts were also elevated upon hyperthermia treatment.
Yunoki et al., 2016 [47]	GSE75127	Identify gene networks involved in the enhancement of hyperthermia sensitivity by the knockdown of BAG3 in human oral squamous cell carcinoma cells	Upregulated: Cell growth and proliferation Post-translational modification Protein folding Downregulated: Cell cycle Gene expression Protein synthesis.	Genes associated with HSPs, such as DNAJB1, HSPA1A, HSPA5, HSPB1, HSPD1, and HSPH1, as well as BAG3 and clusterin (CLU), were up-regulated.

**Table 2 cancers-15-00113-t002:** Comparison of studies and datasets included in the analysis.

		Experimental Parameters							Transcriptome Alterations		
Dataset	Cell Line and Origin	Technique/Platform	Heating Technique	Temp. (°C)	Heating Time (min)	CEM43	Timepoint (Hours)		Total Transcripts	Upregulated (%)	Downregulated (%)
Tabuchi, 2008	Lymphoma (U937)	cDNA Microarray. Human Genome U133A array	Conductive heating: Water bath	41	30	1875	3		12,815	2970 (23.18%)	1708 (13.3%)
Furusawa, 2011 (a)	Lymphoma (U937)	cDNA Microarray. Human Genome U133-plus 2.0	Conductive heating: Water bath	42	90	22.5	0		20,594	5466 (26.54%)	5992 (29.10%)
Furusawa, 2011 (b)	Lymphoma (U937)	cDNA Microarray. Human Genome U133A array	Conductive heating: Water bath	42	15	3.75	1		20,594	4613 (22.40%)	7199 (39.69%)
Furusawa, 2011 (c)	Lymphoma (U937)	cDNA Microarray. Human Genome U133-plus 2.0	Conductive heating: Water bath	42	15	3.75	3		20,594	3084 (14.98%)	13,003 (63.14%)
Furusawa, 2011 (d)	Lymphoma (U937)	cDNA Microarray. Human Genome U133A array	Conductive heating: Water bath	42	15	3.75	6		20,594	3772 (18.32%)	11,589 (56.27%)
Furusawa, 2011 (e)	Lymphoma (U937)	cDNA Microarray. Human Genome U133-plus 2.0	Conductive heating: Water bath	44	15	30	0		20,594	6008 (29.17%)	4770 (23.16%)
Furusawa, 2011 (f)	Lymphoma (U937)	cDNA Microarray. Human Genome U133A array	Conductive heating: Water bath	44	15	30	1		20,594	8779 (42.63%)	3190 (15.49%)
Furusawa, 2011 (g)	Lymphoma (U937)	cDNA Microarray. Human Genome U133-plus 2.0	Conductive heating: Water bath	44	15	30	3		20,594	7611 (36.96%)	5155 (25.03%)
Furusawa, 2011 (h)	Lymphoma (U937)	cDNA Microarray. Human Genome U133A array	Conductive heating: Water bath	44	15	30	6		20,594	7017 (34.07%)	5155 (25.03%)
Amaya, 2014 (a)	Breast cancer (MCF7)	cDNA Microarray. Illumina HumanHT-12 V4.0	Conductive heating: Water bath	45	30	120	4		20,909	1137 (5.44%)	1190 (5.69%)
Amaya, 2014 (b)	Breast cancer (MDA-MB-231)	cDNA Microarray. Illumina HumanHT-12 V4.0 beadchip	Conductive heating: Water bath	45	30	120	4		20,909	1336 (6.39%)	16,385 (78.36%)
Amaya, 2014 (c)	Breast cancer (MDA-MB-468)	cDNA Microarray. Illumina Human HT-12 V4.0 beadchip	Conductive heating: Water bath	45	30	120	4		20,909	132 (0.63%)	1929 (9.23%)
Court, 2017	Ovarian cancer (HeyA8)	cDNA Microarray. Human HT-12 v4 Beadchip	Electromagnetic: Magnetic fluid hyperthermia	43	30	30	0		18,299	0 (0%)	1 (0.01%)
Andocs, 2015 (a)	Colorectal cancer (HT-29)	cDNA Microarray. HGU133 Plus 2.0 arrays	Electromagnetic: Modulated electrohyperthermia	42	30	7.5	4		12,372	8 (0.06%)	1 (<0.01%)
Andocs, 2015 (b)	Colorectal cancer (HT-29)	cDNA Microarray. HGU133 Plus 2.0 arrays	Electromagnetic: Modulated electrohyperthermia	42	30	7.5	24		12,232	0 (0%)	1 (<0.01%)
Yunoki, 2016	Oral squamous cell carcinoma (HSC-3)	cDNA Microarray. Human Genome U133-plus 2.0 array	Conductive heating: Water bath	44	90	180	24		12,815	89 (0.69%)	23 (0.18%)
Scutigliani, 2022 (a) *	Bladder cancer (T24)	RNA sequencing. Illumina Novaseq 6000, paired-end, read depth of 40M.	Conductive heating: Water bath	42	60	15	6		24,418	2370 (9.71%)	2112 (8.65%)
Scutigliani, 2022 (b) *	Cervical cancer (HeLa)	RNA sequencing. Illumina Novaseq 6000, paired-end, read depth of 40 M.	Conductive heating: Water bath	42	60	15	24		17,673	71 (0.40%)	126 (0.71%)

* Unpublished data sets.

## Data Availability

The data presented in this study are openly available in interactive form via figlinq.com (https://create.figlinq.com/dashboard/e.m.scutigliani:1055, accessed on 16 November 2022). All code used for the data analysis is available via GitHub (https://github.com/Krawczyk-Group/Scutigliani-et-al-2022, accessed on 16 November 2022).

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
