# Peer review of "The Effects of Heat Stress on the Transcriptome of Human Cancer Cells: A Meta-Analysis"

_cancers, 2022, doi:10.3390/cancers15010113_

Round 1

Reviewer 1 Report

Although it is very interesting to know how the cells respond to the hyperthermia, I think that it would be interesting to add how the head is given to them and over all in what conditions. It is not the same to have a high temperature exposition, than to obtain it under a spatial gradient or time impuse. 

Reviewer 2 Report

The article deals with the vital topic of oncologic hyperthermia. The meta-analysis of heat stress effect on the transcriptome of human cancer cell lines is gap-filling work, noteworthy information for the emerging clinical applications of hyperthermia.

The basic message of the article is that it found a considerable discrepancy between the transcriptome data of the various published articles, so a unified picture could not be formed. The authors hypothesize that different treatment conditions and cells as well as the various treatment methods lead to the high variation of the results, even though the temperature remains in the hyperthermia range. The practical human application cannot make homogeneous heating and has not only malignant cells in the heated target, but the in vitro cell line samples homogeneous heating conditions, and no different cells mix the information. Despite the definite thermal conditions and homogeneous cell cultures, remarkable differences appear in the analyzed results.

The article's message is essential, but several shortcomings and errors must be corrected in this work before its publication.

General remarks

1.      The variation of the hyperthermia-induced transcriptome of cells treated with different methods is not surprising in light of the differences in the experimental results of various heating methods published earlier. The thermal reaction to the same temperature load measured by the Arrhenius plot varies by the treatment method [[i]], and the rate and kind of cell death also have drastic alterations, measured by flow cytometry [[ii]]. The methods' differences were well followed by comparing two referred articles, [[iii]] and [[iv]]. Furusawa et al. observed no changes at , while Andocs et al. measured significant variation. Both references were published from the same research group and the same U937 cell line. Mention these important pieces of information may complete the submission.

2.   The application of the  as thermal dose follows the clinically applied dose. The conditions are optimal during in vitro experiments because this dose characterizes the entire sample without inhomogeneities (). The transcriptome variation supports the emerging opinions that the  dose does not provide a definite characterization of hyperthermia in clinical applications, as shown in soft tissue sarcomas [[v]]. The administered dose of  for local hyperthermia did not exhibit an association between dose and clinical outcomes [[vi]], while in surface heating, the  is satisfactory [[vii]]. These complications led to thinking about another dose. A new dose for human uterus cervix treatment was introduced: the  dose parameter [[viii]].  correlates with successful (CR) therapy. Therefore I think it is necessary to mention in the maniscript that the  was not satisfactory for dose characterization like we had a sign of this in human clinical practice. This remark touches on the future of clinical hyperthermia [[ix]].

3.      The comparison of the in vitro treatment with clinical therapies has a serious shortcoming. The dose in clinical applications never reaches such a high  dose, as shown in some experiments. The median dose in the relatively easy heating superficial tumors <15 min  [[x]], and even whole-body hyperthermia can not go over 50 min  because the physiological limit () of this therapy. How the signature of stresses depend on the dose?

Correction proposals (following the rows’ numbering, referring to rows)

General error: the references at the end of the manuscript have numeration, while the article massively uses another reference method (name and date). The reader is not able to identify the correct reference. It has to be unified according to the Journal's requirements. What does the letter mean after the date in references (as it shown in (Scutigliani 2022b))?

32. I can not identify the two new datasets which have not been published yet. No such reference is given.

106. Table 1. Reference name Furusawa cut by a new line. It has to be corrected

185-188. How was made the normalization of the data? Was it supposed that the intensity of up and down-regulation of genes is proportional to the dose? It is not a trivial assumption.

199-200. What are the key experimental parameters? The authors show in the text only one (the CEM dose).

204. The declared cluster numbers contradict the numbers in Fig. 1F. This mixture happens multiple times in the submission. Please correct.

208. Fig. 1D. The distribution of the principal component has to be not cumulative because the main massage, how the principal components dominate, does not go through.

208. Fig 1E. (1) The axes need negative signs under zero.

208. Fig.1E. (2) It is not understandable to me how the same PC2 is only positive in the left panel while it has negative values in the middle panel. Are these different PC2s?

208. Fig. 1E. (3) The timing and doses are also different in the panels according to the right panel denotation. When these two PC2s are different, how were they chosen?

219. Table 2. Reference name Furusawa cut by a new line. It has to be corrected.

248. Fig. 2B. What are the blue lines connecting the two clusters?

248. Fig.2F. Is the orange column general cell death or especially apoptosis, as it is written in row 242?

345. Again: what are the unpublished data?

My opinion:

The manuscript gives important info for oncologic hyperthermia. After corrections, I propose its publication.

[[i]] Whitney J, Carswell W and Rylander N: Arrhenius parameter determination as a function of heating method and cellular microenvironment based on spatial cell viability analysis. Int J Hyperthermia 29: 281-295, 2013.

[[ii]]   Yang K-L, Huang C-C, Chi M-S, Chiang H-C, Wang Y-S, Andocs G, et.al. (2016) In vitro comparison of conventional hyperthermia and modulated electro-hyperthermia, Oncotarget, 7(51): 84082-84092, doi: 10.18632/oncotarget.11444, http://www.ncbi.nlm.nih.gov/pubmed/27556507

[[iii]] Furusawa, Y.; Tabuchi, Y.; Wada, S.; Takasaki, I.; Ohtsuka, K.; Kondo, T. Identification of Biological Functions and Gene 495 Networks Regulated by Heat Stress in U937 Human Lymphoma Cells. Int. J. Mol. Med. 2011, 28, 143–151

[[iv]] Andocs, G.; Rehman, M.U.; Zhao, Q.-L.; Tabuchi, Y.; Kanamori, M.; Kondo, T. Comparison of Biological Effects of Modulated 491 Electro-Hyperthermia and Conventional Heat Treatment in Human Lymphoma U937 Cells. Cell Death Discov 2016, 2, 16039

[[v]] Maguire PD, Samulski TV, Prosnitz LR, Jones EL, Rosner GL, Powers B, Layfield LW, Brizel DM, Scully SP, Harrelson JM, et al: A phase II trial testing the thermal dose parameter CEM43 degrees T90 as a predictor of response in soft tissue sarcomas treated with pre-operative thermoradiotherapy. Int J Hyperthermia 17: 283-290, 2001

[[vi]] de Bruijne M, van der Holt B, van Rhoon GC and van der Zee J: Evaluation of CEM43 degrees CT90 thermal dose in superficial hyperthermia: A retrospective analysis. Strahlenther Onkol 186: 436-443, 2010

[[vii]] Dewhirst MW, Vujaskovic Z, Jones E and Thrall D: Re-setting the biologic rationale for thermal therapy. Int J Hyperthermia 21: 779-790, 2005

[[viii]] Franckena M, Fatehi D, de Bruijne M, Canters RA, van Norden Y, Mens JW, van Rhoon GC and van der Zee J: Hyperthermia dose-effect relationship in 420 patients with cervical cancer treated with combined radiotherapy and hyperthermia. Eur J Cancer 45: 1969-1978, 2009

[[ix]] Jones E, Thrall D, Dewhirst MW and Vujaskovic Z: Prospective thermal dosimetry: The key to hyperthermia's future. Int J Hyperthermia 22: 247-253, 2006

[[x]] Jones EL, Oleson JR, Prosnith LR, et.al. (2005) Randomized trial of hyperthermia and radiation for superficial tumors, J Clin Oncol, 23(13): 3079-3085.

Author Response

Please se

Dear Editor,

We are grateful for the favorable and insightful comments of reviewer 2. Below please find our detailed responses to her/his critiques and requests. We are looking forward to the next step in the submission procedure.

Thank you,

on behalf of all authors,
Fernando Lobo

General remarks

  1. The variation of the hyperthermia-induced transcriptome of cells treated with different methods is not surprising in light of the differences in the experimental results of various heating methods published earlier. The thermal reaction to the same temperature load measured by the Arrhenius plot varies by the treatment method [[i]], and the rate and kind of cell death also have drastic alterations, measured by flow cytometry [[ii]]. The methods' differences were well followed by comparing two referred articles, [[iii]] and [[iv]]. Furusawa et al. observed no changes at , while Andocs et al. measured significant variation. Both references were published from the same research group and the same U937 cell line. Mention these important pieces of information may complete the submission.

Thank you for suggesting to consider the heating method as a key experimental parameter. An extra column with this information has been added to Table 2. Important remarks relevant to this parameter were also included to the results section.

  1. The application of the as thermal dose follows the clinically applied dose. The conditions are optimal during in vitro experiments because this dose characterizes the entire sample without inhomogeneities (). The transcriptome variation supports the emerging opinions that the dose does not provide a definite characterization of hyperthermia in clinical applications, as shown in soft tissue sarcomas [[v]]. The administered dose of  for local hyperthermia did not exhibit an association between dose and clinical outcomes [[vi]], while in surface heating, the  is satisfactory [[vii]]. These complications led to thinking about another dose. A new dose for human uterus cervix treatment was introduced: the  dose parameter [[viii]].  correlates with successful (CR) therapy. Therefore I think it is necessary to mention in the maniscript that the  was not satisfactory for dose characterization like we had a sign of this in human clinical practice. This remark touches on the future of clinical hyperthermia [[ix]].

We agree that mentioning the different methods that are used to calculate the thermal dose in the clinic is a valuable addition to the manuscript, and we have made adjustments to the results section.

  1. The comparison of the in vitro treatment with clinical therapies has a serious shortcoming. The dose in clinical applications never reaches such a high dose, as shown in some experiments. The median dose in the relatively easy heating superficial tumors <15 min [[x]], and even whole-body hyperthermia can not go over 50 min  because the physiological limit () of this therapy. How the signature of stresses depend on the dose?

The question whether the gene expression signature of stress response is correlated with the thermal dose is indeed an important one, and among the main reasons for initiating this study. Based on the data that we have analyzed, however, we are not able to establish associations between the thermal dose and transcriptome alterations, likely because of the high degree of variability, driven by the other major differences in experimental approaches used in the analyzed studies. We already stress this point a number of times throughout the manuscript.

Correction proposals (following the rows’ numbering, referring to rows)

General error: the references at the end of the manuscript have numeration, while the article massively uses another reference method (name and date). The reader is not able to identify the correct reference. It has to be unified according to the Journal's requirements. What does the letter mean after the date in references (as it shown in (Scutigliani 2022b))?

These name-based references (name, date) allude to the data sets included in the study, and are listed in Table 2. We believe that this is a general practice in the field and prefer to adhere to it for general readability, rather than converting the references to numerical.

Line 32. I can not identify the two new datasets which have not been published yet. No such reference is given.

We added a remark on Table 2 to clarify that these are our own unpublished data sets, Scutigliani 2022a and Scutigliani 2022b. The procedures for generating these datasets are already part of the Material and Methods section.

Line 106. Table 1. Reference name Furusawa cut by a new line. It has to be corrected

We thank the reviewer for this remark and we will make sure to forward this issue to the publisher. In addition, Table 1 has been adjusted.

Line 185-188. How was made the normalization of the data? Was it supposed that the intensity of up and down-regulation of genes is proportional to the dose? It is not a trivial assumption.

The magnitude of gene expression was not normalized, but included as reported in original studies. We considered genes as up- or down-regulated if their expression changed at least 1.5 fold. In our opinion, there is no simple biological relationship between the heat dose and gene expression that could be used for normalization.

Line 199-200. What are the key experimental parameters? The authors show in the text only one (the CEM dose).

The following key experimental parameters are described on Table 2: Cell line, Transcriptomic Technique, Heating Technique, Temperature, Heating Time, CEM43, and Time point of transcriptomic. We decided not to mention them individually in line 199 for the sake of text readability, but we did add a reference to Table 2 for all key experimental parameters (line 199).

Line 204. The declared cluster numbers contradict the numbers in Fig. 1F. This mixture happens multiple times in the submission. Please correct.

Thank you for critically evaluating the figures, the misplaced annotations have been corrected.

Line 208. Fig. 1D. The distribution of the principal component has to be not cumulative because the main massage, how the principal components dominate, does not go through.

We agree with the reviewer that the visualization of individual component contributions creates a different perspective of the data. In this particular figure, however, we chose components 1-3 because together they explain >50% of the variance in the data, and we wanted to visualize this in figure 1d as a cumulative plot. To indicate this intention, we have added a line marking 50% of total variance and marked the chosen PCs in a different color.

Line 208. Fig 1E. (1) The axes need negative signs under zero.

Thank you for critically evaluating the figures, negative signs were indeed accidentally removed when exporting figures and were now re-added.

Line 208. Fig.1E. (2) It is not understandable to me how the same PC2 is only positive in the left panel while it has negative values in the middle panel. Are these different PC2s?

The PC2 has negative and positive values on both graphs in Fig. 1E, the corresponding negative signs were added (also see our reply to the previous comment).

Line 208. Fig. 1E. (3) The timing and doses are also different in the panels according to the right panel denotation. When these two PC2s are different, how were they chosen?

The colors were adjusted for better visualization and correspondence.

Line 219. Table 2. Reference name Furusawa cut by a new line. It has to be corrected.

We will make sure to forward this issue to the publisher, thank you for your remark.

Line 248. Fig. 2B. What are the blue lines connecting the two clusters?

The blue boxes with blue lines are box plots and their corresponding whiskers. We have increased the size of the error bar whiskers to avoid confusion.

Line 248. Fig.2F. Is the orange column general cell death or especially apoptosis, as it is written in row 242?

The orange column is referring, indeed, to general cell death.

Line 345. Again: what are the unpublished data?

We added a remark on Table 2 to clarify that these are our own unpublished data sets, Scutigliani 2022a and Scutigliani 2022b.

My opinion:

The manuscript gives important info for oncologic hyperthermia. After corrections, I propose its publication.

[[i]] Whitney J, Carswell W and Rylander N: Arrhenius parameter determination as a function of heating method and cellular microenvironment based on spatial cell viability analysis. Int J Hyperthermia 29: 281-295, 2013.

[[ii]]   Yang K-L, Huang C-C, Chi M-S, Chiang H-C, Wang Y-S, Andocs G, et.al. (2016) In vitro comparison of conventional hyperthermia and modulated electro-hyperthermia, Oncotarget, 7(51): 84082-84092, doi: 10.18632/oncotarget.11444, http://www.ncbi.nlm.nih.gov/pubmed/27556507

[[iii]] Furusawa, Y.; Tabuchi, Y.; Wada, S.; Takasaki, I.; Ohtsuka, K.; Kondo, T. Identification of Biological Functions and Gene 495 Networks Regulated by Heat Stress in U937 Human Lymphoma Cells. Int. J. Mol. Med. 2011, 28, 143–151

[[iv]] Andocs, G.; Rehman, M.U.; Zhao, Q.-L.; Tabuchi, Y.; Kanamori, M.; Kondo, T. Comparison of Biological Effects of Modulated 491 Electro-Hyperthermia and Conventional Heat Treatment in Human Lymphoma U937 Cells. Cell Death Discov 2016, 2, 16039

[[v]] Maguire PD, Samulski TV, Prosnitz LR, Jones EL, Rosner GL, Powers B, Layfield LW, Brizel DM, Scully SP, Harrelson JM, et al: A phase II trial testing the thermal dose parameter CEM43 degrees T90 as a predictor of response in soft tissue sarcomas treated with pre-operative thermoradiotherapy. Int J Hyperthermia 17: 283-290, 2001

[[vi]] de Bruijne M, van der Holt B, van Rhoon GC and van der Zee J: Evaluation of CEM43 degrees CT90 thermal dose in superficial hyperthermia: A retrospective analysis. Strahlenther Onkol 186: 436-443, 2010

[[vii]] Dewhirst MW, Vujaskovic Z, Jones E and Thrall D: Re-setting the biologic rationale for thermal therapy. Int J Hyperthermia 21: 779-790, 2005

[[viii]] Franckena M, Fatehi D, de Bruijne M, Canters RA, van Norden Y, Mens JW, van Rhoon GC and van der Zee J: Hyperthermia dose-effect relationship in 420 patients with cervical cancer treated with combined radiotherapy and hyperthermia. Eur J Cancer 45: 1969-1978, 2009

[[ix]] Jones E, Thrall D, Dewhirst MW and Vujaskovic Z: Prospective thermal dosimetry: The key to hyperthermia's future. Int J Hyperthermia 22: 247-253, 2006

[[x]] Jones EL, Oleson JR, Prosnith LR, et.al. (2005) Randomized trial of hyperthermia and radiation for superficial tumors, J Clin Oncol, 23(13): 3079-3085.

e the attachment
